# Wave Propagation Characteristics and Compaction Status of Subgrade during Vibratory Compaction

**DOI:** 10.3390/s23042183

**Published:** 2023-02-15

**Authors:** Junkai Yao, Mao Yue, Hongsheng Ma, Changwei Yang

**Affiliations:** 1State Key Laboratory for Track Technology of High-Speed Railway, China Academy of Railway Sciences Corporation Limited, Beijing 100081, China; 2MOE Key Laboratory of High-Speed Railway Engineering, College of Civil Engineering, Southwest Jiaotong University, Chengdu 610031, China; 3Sichuan Highway Planning Survey Design and Research Institute, Chengdu 610041, China

**Keywords:** subgrade engineering, vibratory compaction, wave propagation coefficient, dynamic modulus of deformation

## Abstract

Vibratory compaction status has a significant influence on the construction quality of subgrade engineering. This study carried out field experiments to study the propagation characteristics of the vertical vibration wave in the soil field along the traveling direction of the vibratory roller. The propagation coefficients of the peak acceleration at different positions and compacting rounds are compared in both the time and frequency domains. The compaction status is estimated in the form of dynamic modulus of deformation (*E*_vd_) obtained by plate load tests. The experiment results show that the propagation coefficient of peak acceleration is affected by the traveling speed, excitation amplitude, and frequency of the vibratory roller, as well as the compacting rounds. An exponential relationship between the wave amplitudes of the fundamental mode and higher-order modes is revealed. The amplitude of the fundamental wave is maximum at the speed of 3 km/h, whereas the amplitudes of higher-order waves have a maximum of 1.5 km/h. The influences of compaction rounds on the average value of *E*_vd_ are also investigated to provide a practical reference for engineering construction.

## 1. Introduction

As an extremely important part of road construction, the compaction of roadbeds has been widely studied by many scholars, where the unevenness of compaction is one of the important factors leading to uneven pavement. In order to solve this problem, the California Department of Highways introduced the California Bearing Ratio (CBR) in the 20th century and applied it to the corresponding compaction specifications [1]. Proctor introduced a novel method of soil compaction and applied it for the first time to the construction of the Bouquet Canyon Reservoir in Los Angeles, USA, which was well received [2]. After this, intelligent compaction (IC) technology, which incorporates several functions, was first introduced in Europe in the late 1970s and is now also gradually being used in practical projects because of its advantages such as enabling real-time monitoring and automatic adjustment of the compaction process. Subsequently, many scholars have proposed different compaction quality testing indicators or systems. Hani, H T. et al. evaluated the construction quality of aggregate base layers based on the variability and uniformity of density and modulus measurements during HMA pavement construction in Wisconsin and assessed the effect of subgrade input parameters on the long-term performance of HMA pavements along with sensitivity analysis [3]. Badakhshan, E., et al. proposed a generalized damage criterion based on uncorrelated flow rules for predicting the stress–strain behavior of unbound granular materials and evaluated the applicability of this model by comparing the predicted results with the actual results [4]. Methods for obtaining the compaction of roadbed fill include the density measurement method and the compaction meter method, where the compaction judgment is made by calculating the relationship between the fundamental and harmonic amplitudes of acceleration signals. Although a lot of theories are available now, theories still need to be verified through practical tests [5,6,7,8]. The compaction meter method is done by calculating the relationship between the amplitude of the acceleration signal of the fundamental wave and the harmonic wave, which is a process from damage measurement to non-damage measurement, and also a process from non-real-time monitoring to real-time monitoring proposed a dynamic method—vibration compacting value to monitor the compaction of subgrade in real-time [9,10,11,12,13]. Imran, S A. et al. proposed a real-time monitoring system for rolling quality based on an artificial neural network [14]. Given the situation that CMV cannot better reflect the compaction quality in some situations, Ma, Y. et al. proposed an intelligent compaction quality evaluation index—accelerated intelligent compaction value (AICV) [15]. Ma, Y. et al. verified the accuracy of the calculation results of intelligent subgrade soil compaction through numerical simulation [16]. Lin, D. obtained the relation between compaction degree, amplitude, and damping ratio of vibration wave through spectrum analysis of acceleration signal through drop-hammer testing [17]. Through MATLAB simulation calculation and other methods, Liu, H.H. et al. concluded that the optimal compaction speed of the vibratory roller was 4 km/h when it met the code requirements of 94% relative to the maximum theoretical density [18]. Hou, Z.Y. et al. designed and developed a remote monitoring system with intelligent compaction index CMV as the core [19]. Wang, S.X. obtained the optimal compaction speed matching with the vibration frequency in the construction process by field test [20]. Ye, Y.S. et al. studied the attenuation coefficient of peak acceleration in the horizontal direction of an adjacent lane by field test [21]. Jia, M.C. et al. proposed the PFC/FLAC coupling method to study the process of dynamic densification of granular soils [22]. Dan, H.C. et al. used sensors to measure the dynamic response relationship between the vibrating wheel and the filler during the rolling process [23].

Zhang, Z.P. et al. proposed a method for measuring the degree of layered compaction of loess roadbed based on hydraulic compaction given the theory of layered compaction [24]. Liu, L. et al. proposed a parameter adjustment strategy for the frequency and amplitude of the drum, based on the coupled nonlinear dynamic model of the vibration wheel [25]. Chen, A.J. et al. studied the propagation and attenuation laws of vibration waves on the ground by conducting field tests [26]. Yang, C.W. et al. conducted a field test to study the dynamic characteristics of the filling during the filling process [27].

The abovementioned research focuses on various aspects of compaction. In the actual compaction process, the dynamic response characteristics of the vibration waves in the horizontal direction of the filler are also very important, but there are still relatively few papers on this aspect. By studying the diffusion law of the vibration wave in the horizontal direction, we can obtain information on the attenuation degree of vibration energy, to study the influence of the change of parameters such as travel speed and vibration frequency on the compaction of the filler to improve the compaction efficiency and quality. Based on this, this study conducted a field prototype test because of the shortage of existing research, obtained the acceleration time range data at different monitoring points during the compaction process by burying acceleration sensors on the surface of the filler, and analyzed the peak propagation coefficient, time-history curve, and spectral characteristics of the filler surface to study the peak acceleration coefficient and optimum compaction conditions of the filler at different speeds of vibratory rollers. The peak acceleration propagation coefficients and the optimum compaction conditions were investigated.

## 2. Test Overview

In this paper, the actual construction line is used as the experimental section for the vibration compaction test. The site soil is sandy with a moisture content of 3.47%, and the particle gradation curve is shown in Figure 1.

### 2.1. Test Equipment

The vibratory roller adopted in this test is a single-drum roller with two types of working modes: strong excitation and weak excitation. The mechanical parameters of used vibration roller in the test are listed in Table 1. DH5922D, a 32−channel dynamic data acquisition equipment of the Donghua test, and 1C303 acceleration sensor of the Donghua test were used in the test, with a range of ±16 g.

Since the test site is located outside and is often rainy, a layer of glass glue is applied around the sensor to make it waterproof and protect the sensors. Steel pipes are embedded in the connection line between the sensors and the data acquisition system to protect the safety of the line. Since the sensors are three-way sensors, the forward direction of the vibratory roller is the X−direction. The direction perpendicular to the advancing direction of the vibratory roller and parallel to the subgrade surface is the Y−direction. The direction perpendicular to the subgrade surface and parallel to the vibration direction of the vibratory roller is the Z−direction. To ensure that the sensors did not shift in all directions during the experiments, a square thin plate with lightweight and high stiffness was pasted at the bottom of the sensor, as shown in Figure 2.

### 2.2. Sensor Layout

To analyze the propagation mechanism of vibration waves in the filling, the sensors are arranged in the filling soil in the way of filling and embedding. Six acceleration sensors were arranged along the advancing direction of the single−lane vibratory roller in the test. To ensure the accuracy of acceleration data, the acceleration sensors were buried in the middle of the lane, and the buried depth was 5 cm below the surface of the filling soil layer. A plate load test is carried out between every two acceleration measuring points to detect the compaction performance of the subgrade filling soil after each compaction, as shown in Figure 2c. To test the compaction quality of the roadbed, the dynamic deformation modulus index is commonly used for evaluation. The dynamic deformation modulus (*E*_vd_) is obtained from a plate test with a certain size and time of action load applied by a falling hammer impact, which represents the ratio of dynamic stress to dynamic strain at a point in the roadbed and describes the ability of the point to resist the dynamic deformation generated by the dynamic load in a certain state.

### 2.3. Test Procedures

Six test lanes are designed in the area of uniform and flat filling for the continuous compaction test of the subgrade. The compaction process parameters of each lane are shown in Table 2. The specific test steps are as follows.

(1)Before the start of the test, the moisture content and particle grading of the filling should be first detected.(2)After the static pressure leveling of each test area by vibratory roller, the rolling compaction is carried out according to the corresponding test conditions. A round compaction process includes two phases, rolling travel from measuring point 1 to 6, and rolling travel from measuring point 6 to 1. After each round of rolling tests is completed, the *E*_vd_ test shall be conducted in the rolled area.(3)Six acceleration measuring points are arranged on the ground with a distance of 2 m between adjacent measuring points to collect the acceleration response of the filling. The field between adjacent measuring points is divided into 5 sections on average, with each section having a width of 0.4 m. As shown in Figure 3, select the middle three sections for the *E*_vd_ test, and three tests are conducted on each section.(4)Synchronously collect the acceleration response of the ground and the vibratory roller, and ensure that the time between them is consistent.

## 3. Test Results and Analysis

In order to study the compaction state and acceleration propagation law of the filling soil under different vibration parameters of the roller, we designed tests under different parameters in the working process of vibratory rollers and used one round as a compaction test to obtain the acceleration time-history curve under different working conditions. For acceleration time-course data, the method of processing the data is quite important [28,29]. The sampling frequency is the number of test data collected by the acceleration sensor per second. If the sampling frequency is too low, the acceleration curve will have obvious burrs, which will reduce the accuracy of the experiment [30]. In order to ensure the smooth curve and the accuracy of the test, the sampling frequency in this paper is 5000 Hz, that is, the sampling interval is 0.0002 s. In this article, the acceleration time-history curve of Z−direction and acceleration of 0.5 s are selected, and the detailed display of the time−history curve is shown in Figure 4.

### 3.1. Propagation of Peak Acceleration

In the working process of the vibratory roller, the vibratory roller not only has an impact on the vibrating wheel and the working area under it but also has a certain impact on the surrounding filling soil. In this paper, the propagation law of vibration wave along the traveling direction is analyzed by taking the peak acceleration of each acceleration measuring point as the research object. Figure 5 shows the acceleration time−history curve of a compaction process. According to the analysis, the effective vibration process is about 35 s. The vibratory roller first goes through measuring point 1, then passes through measuring points 2, 3, 4, and 5 in sequence, and finally to measuring point 6. Then it returns from point 6 to point 1 to complete a set of vibrating compaction tests.

The acceleration peak at each measurement point can be studied by obtaining the data from the acceleration sensors buried at six measurement points on the surface of the fill, and then the acceleration peak propagation coefficient at the remaining measurement points can be studied when the vibratory roller passes through a certain measurement point, as shown in Figure 6. Analysis of Figure 6 shows that the acceleration peak value decreases significantly along the direction of advance of the vibratory roller as the vibration wave in the filler soil.

#### 3.1.1. Propagation Coefficient of Peak Acceleration under Different Compaction Conditions

During the propagation of the vibration wave in the filling, due to internal effects such as friction between particles in the subgrade filling, part of the energy of the vibration wave is consumed. The interaction between the particles is closely related to the compactness of the subgrade, so the compaction of the subgrade filling has a certain impact on the attenuation of the vibration wave [28]. In this paper, the ratio of peak acceleration in the same time period is taken as the peak acceleration propagation coefficient (k, k ≤ 1). When the vibratory roller works to measure point 1, the peak acceleration of point 1 is the largest, and the ratio of the peak acceleration value of measuring point 2 to measuring point 6 to measuring point 1 is taken as the peak acceleration propagation coefficient of each measuring point. This method is also used to calculate the propagation coefficient of acceleration when the vibratory roller works at other measuring points.

From the analysis of Figure 7, it can be seen that no matter which working position the vibratory roller is in, the peak acceleration of the soil uncompacted is greater than that in the compacted state. The maximum value of the peak acceleration difference occurs at the working position of the vibratory roller, and the adjacent position is second. As can be shown in Figure 8, when the vibratory roller is running directly above each measuring point, the propagation coefficient of peak acceleration in the uncompacted state is greater than that in the compacted state of adjacent measuring points along the driving direction. The main reason for this phenomenon is that in the test area where the filling has been rolled, the upper soil and the lower soil are tightly bonded together, in a relatively dense state, which has a greater restriction on the sensor, making the sensor can be better embedded in the soil and vibrate together with the soil. The uncompacted soil is relatively fluffy, and the upper soil is not tightly bonded to the lower soil, which has a less restrictive effect on the sensor and enables the sensor to receive greater vibration energy.

According to the analysis, the peak acceleration and propagation coefficient of the vibration wave near the working position of the vibratory roller in the uncompacted state is greater than that of the compacted state in the process of horizontal propagation. It is concluded that the reason why the compaction of soil increases during the working process of the vibratory roller is that the reduction of pores in the vertical direction makes the upper and lower soil more closely connected, which destroys the original bonding effect in the horizontal direction. From the field filling in the compacted test section, it can be seen that there are many transverse penetration cracks in the surface perpendicular to the advanced direction of the vibratory roller. These cracks exist in both strong and weak vibration conditions, and the more rounds of compaction, the more obvious the penetration cracks are. From this phenomenon, it can be seen that with the improvement of the soil compaction state, the damping ratio of soil in the horizontal direction becomes larger and larger, so the propagation coefficient of peak acceleration decreases with the improvement of the soil compaction state.

In the process of vibration wave propagation, the acceleration peak propagation coefficient decreases due to the increase of distance effect and damping effect. With the gradual increase in distance in the horizontal direction, the influence of the distance effect increases, so the increase in the damping effect accounts for the second factor. Therefore, the effect of soil compaction on the peak acceleration propagation coefficient in the working area far away from the vibratory roller is not as good as that in the adjacent working area.

#### 3.1.2. Propagation Coefficient of Peak Acceleration of Vibratory Roller at Different Speeds

Different speeds of the vibratory roller have a certain influence on the propagation law of the vibration wave in the filling. In order to further study the horizontal propagation law and horizontal propagation coefficient of the vibration wave, in this paper, the peak acceleration propagation coefficient is selected under the conditions of strong vibration when the speed of the vibratory roller is 1.5 km/h, 3 km/h and 4.5 km/h, and the filling under different compaction states are as shown in Figure 9.

Figure 9a shows that the soil is in an uncompacted state, and the peak acceleration propagation coefficient has a negative correlation with the speed of the vibratory roller, that is, when the speed of the vibratory roller is 1.5 km/h, the vibration wave propagation coefficient in the horizontal direction is the largest, followed by 1.5 km/h, and the propagation coefficient is the smallest at 4.5 km/h. The analysis shows that when the soil is in the uncompacted state, the slower the speed of the vibratory roller, the better the propagation of the vibration wave along the horizontal direction. That is, the same test area is vibrated and compacted under the same working conditions, and the energy carried by the vibration wave is the same when the vibration wave moves from the vibratory roller to the filling. At this time, the slower the speed of the vibratory roller, the greater the horizontal propagation coefficient of the peak acceleration, and the slower the vibration wave propagates in the same distance, the greater the energy carried by the vibration wave, which is more conducive to soil compaction.

According to Figure 9b, when the soil is in the compacted state, the propagation coefficient of peak acceleration is not negatively correlated with the speed of the vibratory roller as it is in the uncompacted state. Instead, when the speed of the vibratory roller is 3 km/h, the horizontal propagation coefficient of peak acceleration of the soil is greater than that of the vibratory roller whose speed is 1.5 km/h and 4 km/h. The analysis shows that the maximum propagation coefficient of peak acceleration occurs between 1.5 km/h and 4 km/h when the soil is compacted.

When the filling soil is in an uncompacted state, decreasing the speed of the vibratory roller is beneficial to the compaction of the filling soil, and with the increase of vibratory roller compaction rounds, the soil compaction condition becomes better and better. At this time, the optimal speed of the vibratory roller for compaction is not the smaller the better, while the vibratory roller’s optimum working speed should be near 3 km/h after the soil is in a state of compaction. The amplification coefficient of peak acceleration is closely related to material damping, gradation, moisture content, and frequency and amplitude of the vibration wave itself.

#### 3.1.3. The Influence of Different Amplitudes and Frequencies on the Propagation Coefficient of Peak Acceleration during Vibratory Roller Operation

In the compaction process of subgrade, the combination of strong vibration and weak vibration is a common working condition in practical engineering. In past work, more attention was paid to the compaction degree of strong vibration and weak vibration to subgrade soil, and less attention to the upward propagation law of vibration wave in the filling soil with strong vibration and weak vibration. In this paper, the influence of the peak acceleration propagation coefficient under strong and weak vibration conditions is studied.

It can be seen from Figure 10 that the peak acceleration propagation coefficient under the strong vibration condition is always greater than that under the weak vibration condition, whether the filling soil is in a compacted or uncompacted state. The analysis shows that the natural frequency of the soil itself in the compacted state and the uncompacted state is more similar to the frequency of the vibratory roller under strong vibration conditions, so the energy propagation coefficient of the vibration wave is larger during the propagation process of filling soil.

### 3.2. Spectrum Analysis of Acceleration Signal

Fast Fourier transform is a common method for signal processing, in which Fourier forward transform is a processing method to transform time-domain signal into frequency−domain signal.
$$F(w)=\underset{T\to \infty }{lim}\int_{\frac{T}{2}}^{\frac{T}{2}}f\left(t\right)e^{-j\omega t}dt$$

f(t)—Original signal

*T*—Period of the original signal

$e^{-j\omega t}$—Function of complex variable

The original acceleration signals of different compaction rounds are transformed by FFT, and the results are shown in Figure 11. It can be seen from the figure that the frequency of the fundamental wave and the frequency of the harmonic wave in the filling do not change with the increase of the number of compactions, while the amplitude of the vibration wave changes with the increase of the number of compactions, in which the amplitude of the fundamental wave first increases and then decreases. According to the analysis, this is because the degree of compaction does not increase linearly with the increase in the number of compaction rounds, but when the degree of compaction reaches a certain value, it will fluctuate up and down around a certain value if it is pressed again.

The amplitudes of fundamental and harmonic waves are obtained from the original acceleration signal by FFT. As the speed of the vibratory roller has a certain influence on the horizontal direction of the vibration wave, the relationship between the amplitude of the fundamental wave and harmonic wave is further studied by only changing the speed of the vibratory roller under the condition that other variables remain unchanged, as is shown in Figure 11.

The amplitude law of the fundamental wave at different speeds of vibratory rollers is similar to the amplitude law of the fundamental wave at different rounds of compaction and the horizontal propagation coefficient law of the peak value at different speeds of vibratory rollers. That is, with the increase in speed, the amplitude of the fundamental wave does not change monotonously, but when the speed of the vibratory roller is 3 km/h, the amplitude of the fundamental wave is greater than that of the vibratory roller when the speed is 1.5 km/h and 4.5 km/h. This law is consistent with the law of the horizontal propagation coefficient of vibration waves at different speeds. When the speed of the vibratory roller is 3 km/h, the horizontal propagation coefficient of the vibration wave peak is the largest. The amplitude of harmonic waves has a different relationship from that of the fundamental wave. When the speed of the vibratory roller is 1.5 km/h, the amplitude of the harmonic is obviously greater than that of the vibratory roller with speeds of 3 km/h and 4.5 km/h.

From the analysis of Figure 12, it can be seen that the relation between the amplitudes of fundamental waves and multiple harmonic waves is nonlinear, fitting y = a·x^b^, and the specific parameters of the fitting results a and b can also be seen from the figure, in which the square of the similarity coefficient (R^2^) of the fitting curves of fundamental waves and harmonic waves of different vibratory roller speeds and different measuring points is at least 0.97, while the rest R2 are above 0.99.

### 3.3. Plate Load Test

This test is divided into six rounds, each round of compaction including two phases: the vibratory roller travels from measuring point 1 to 6, and then from measuring point 6 to 1. In order to analyze the compaction of subgrade soil after each round of rolling, the traditional test method plate load test is used to determine the *E*_vd_, and finally, the compaction degree of subgrade soil is evaluated.

In this paper, 15 points of each *E*_vd_ measurement are taken as the abscissa, and the number of compactions is taken as the ordinate to draw the contour map measured by *E*_vd_ and the broken-line map measured at different points under different rounds of compactions. According to the contour map, the continuous change in *E*_vd_ along the abscissa direction shows the compaction condition of the test lane after each vibration compaction. From the longitudinal coordinate direction, it can be seen that the continuous change in *E*_vd_ can be obtained by six measurements of the same measuring point. The contour maps show the *E*_vd_ value in different colors, and the *E*_vd_ value corresponding to red and yellow is larger than that of blue. By linear fitting between two adjacent points, the continuous variation in the number of vibration compactions along the test section and different points in the same lane can be obtained. The broken−line chart shows the specific *E*_vd_ of different compaction rounds in the abscissa direction of the contour map.

From the analysis of Figure 13, it can be seen that the contour map corresponding to the vibratory roller under strong vibration conditions at different speeds is dominated by warm colors, while the contour map corresponding to the vibratory roller under weak vibration conditions at different speeds is dominated by cold colors. According to the analysis, the reason for this phenomenon is that the *E*_vd_ value appears when the test condition is in weak vibration and is larger than that of other measuring points, while the *E*_vd_ value appears when the test condition is in strong vibration and is smaller than that of other measuring points.

It can be seen from Figure 14 that the average curve of *E*_vd_ of the first three rounds of compaction is quite different. We think that the main reason for this situation is that the compaction degree of soil has a big difference when it is laid loose, so the last three compacts are taken as the main analysis object. According to the analysis, the mean value of *E*_vd_ under strong vibration conditions and weak vibration conditions does not show a monotonous change with the increase in the number of vibration compactions. When the speed of the vibratory roller is 3 km/h and the vibration condition is weak vibration, the mean value of *E*_vd_ of the sixth compaction is larger, with a magnitude of 28.5 MPa, while the *E*_vd_ mean value of the other conditions is around 23 MPa. This corresponds to the vibratory roller being at 3 km/h, which has the maximum amplitude of the fundamental wave and the maximum propagation coefficient of peak acceleration under the soil compacted state in the previous paper.

## 4. Conclusions

Through field tests of vibration compaction, the following conclusions are drawn.

(1)Due to the damping of soil mass and the joint action of upper soil and lower soil mass, the peak acceleration propagation coefficient in the uncompacted state is greater than that in the compacted state at adjacent measuring points along the driving direction of the vibratory roller.(2)When the speed of the vibratory roller is 3 km/h, the acceleration propagation coefficient is greater than that at 1.5 km/h and 4.5 km/h. The speed of the vibratory roller that is most conducive to the compaction of the test-filling soil is between 1.5 km/h and 4.5 km/h. The specific value needs more detailed design test and analysis.(3)When the speed of the vibratory roller is 3 km/h, the amplitude of the fundamental wave at each acceleration measuring point reaches the maximum, while when the speed of the vibratory roller is 1.5 km/h, the amplitude of the harmonic wave at acceleration measuring point reaches the maximum. The relationship between the amplitude of the fundamental wave and the amplitude of multiple harmonic waves of each vibration wave is y = a·x^b^.(4)Through several rolling tests, it was found that the average *E*_vd_ value of the vibratory roller under the weak vibration and the speed of 3km/h was significantly higher than the rest of the working conditions.

## Figures and Tables

**Figure 1 sensors-23-02183-f001:**
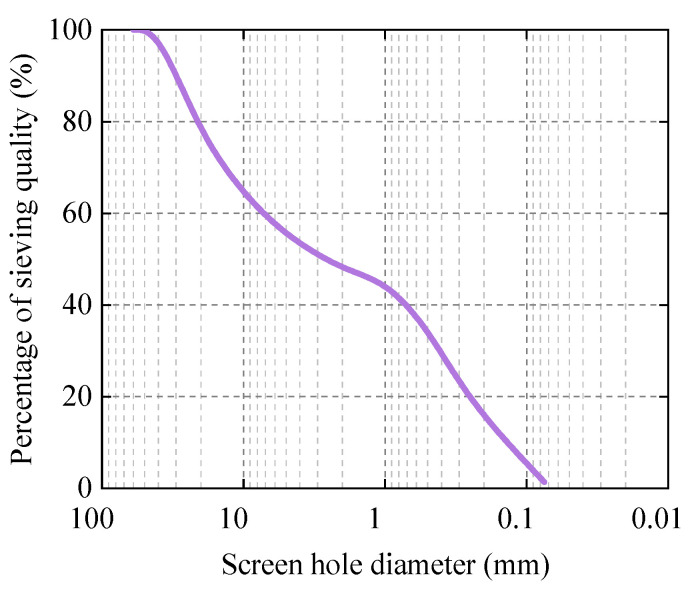
Particle grading curve of filling soil.

**Figure 2 sensors-23-02183-f002:**
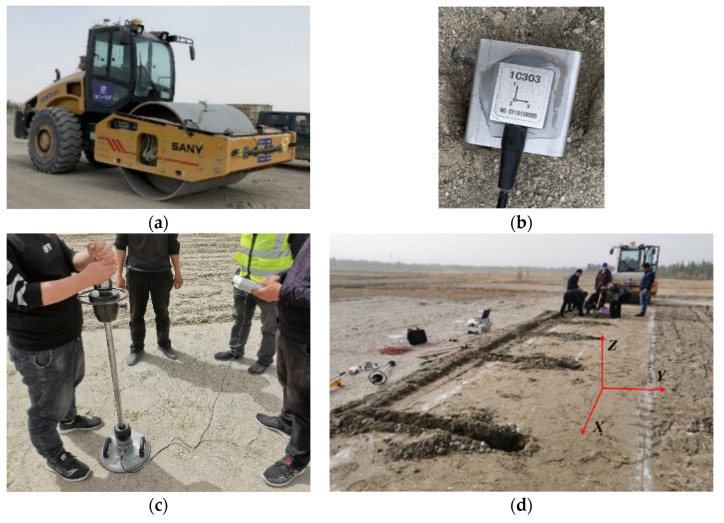
Test site and test equipment. (**a**) Vibratory roller. (**b**) Accelerator sensor. (**c**) Plate load test. (**d**) Test site.

**Figure 3 sensors-23-02183-f003:**
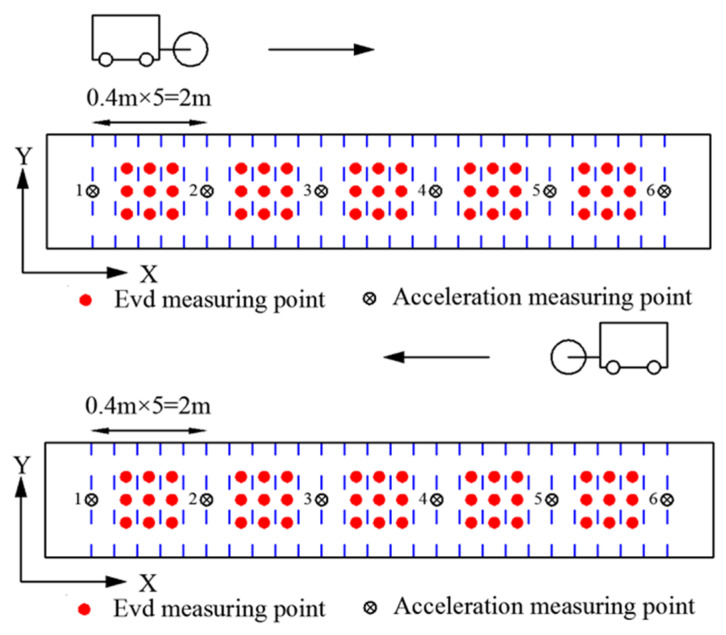
A round compaction process.

**Figure 4 sensors-23-02183-f004:**
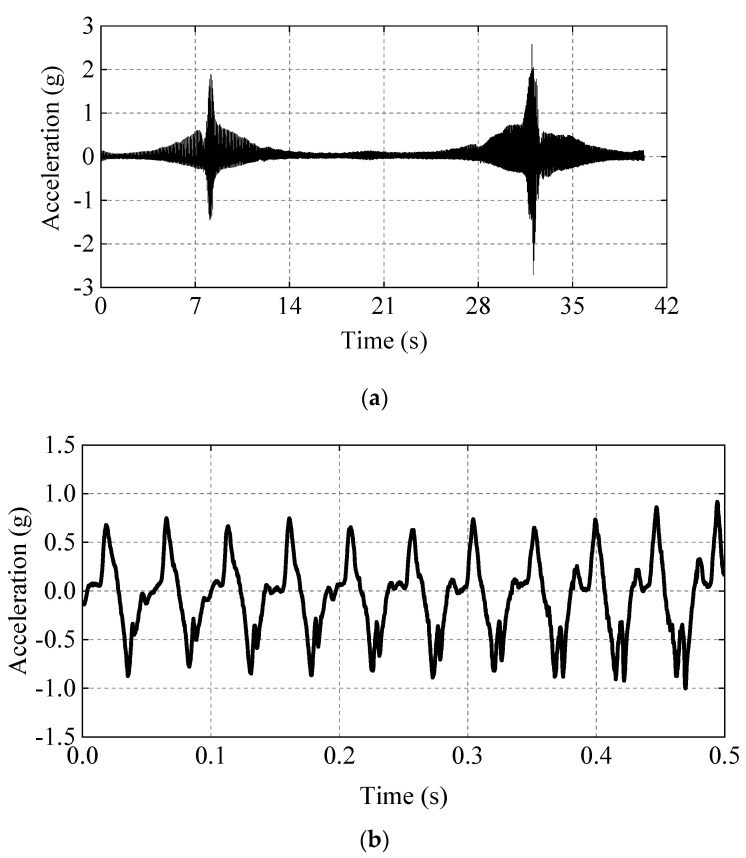
Time−history curve of acceleration. (**a**) Acceleration time−history curve in Z−direction (The whole time period). (**b**) Acceleration time−history curve in Z-direction (The partial time period).

**Figure 5 sensors-23-02183-f005:**
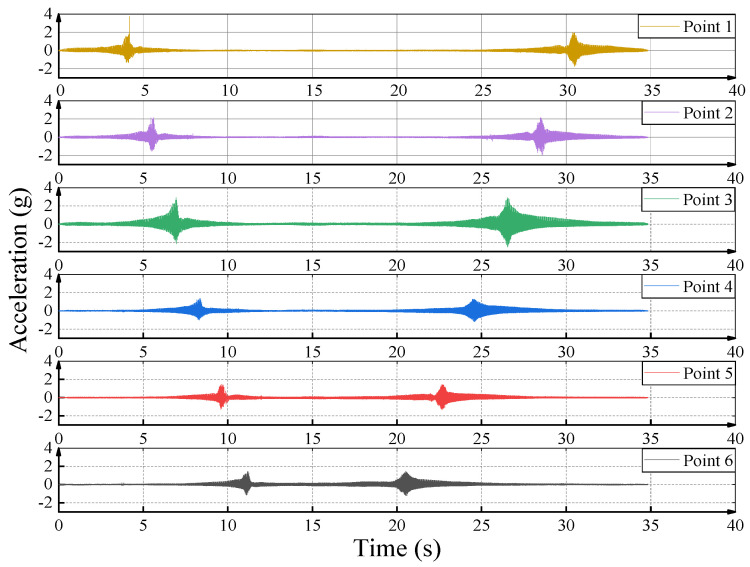
Time−history curve of accelerations from point 1 to point 6.

**Figure 6 sensors-23-02183-f006:**
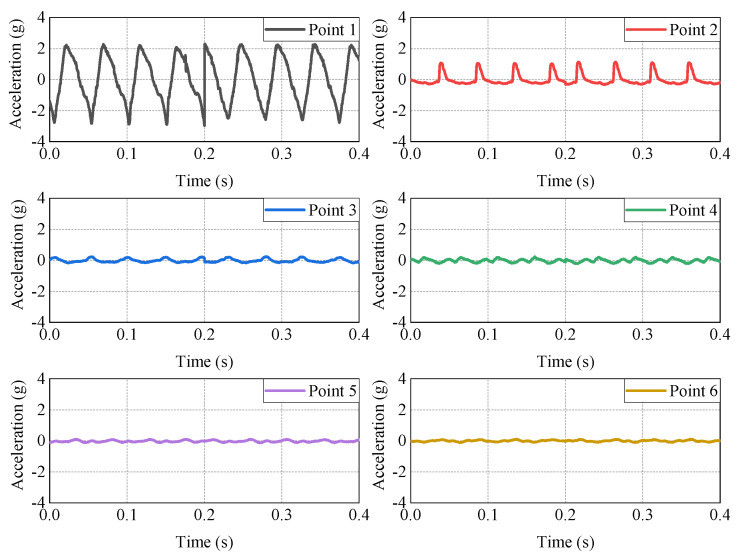
The time−history curve of each measuring point.

**Figure 7 sensors-23-02183-f007:**
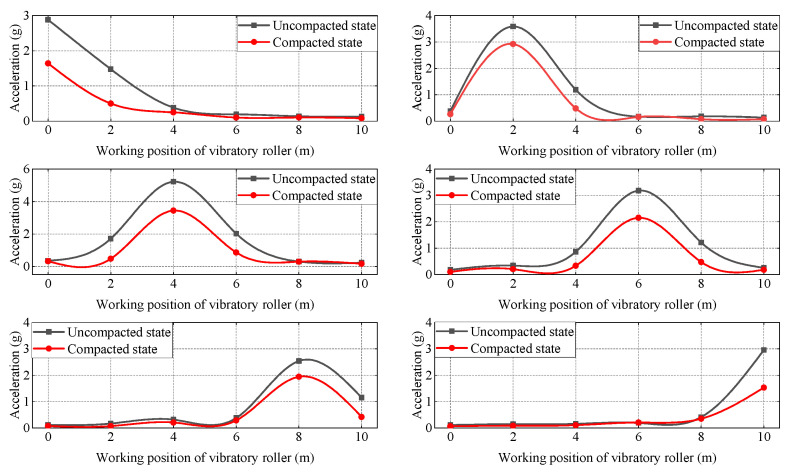
Peak acceleration attenuation curve.

**Figure 8 sensors-23-02183-f008:**
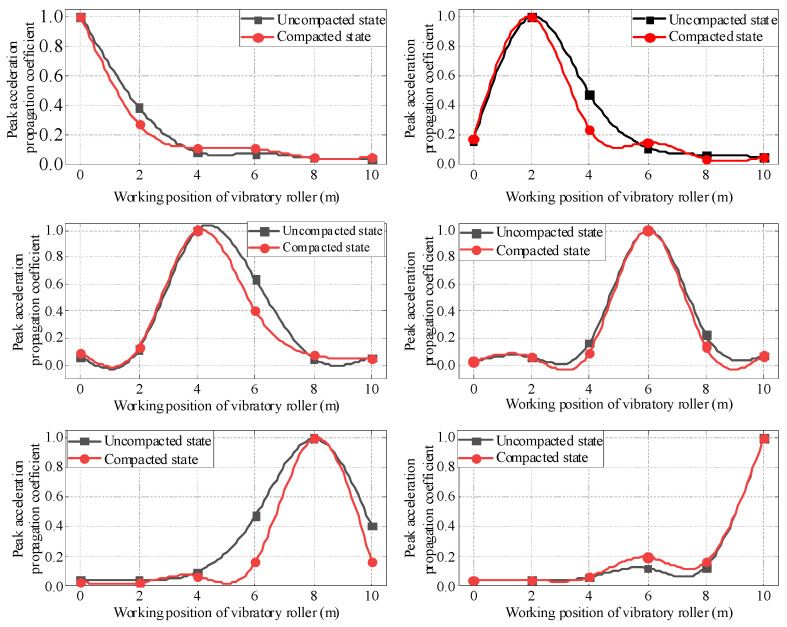
Peak acceleration propagation coefficient.

**Figure 9 sensors-23-02183-f009:**
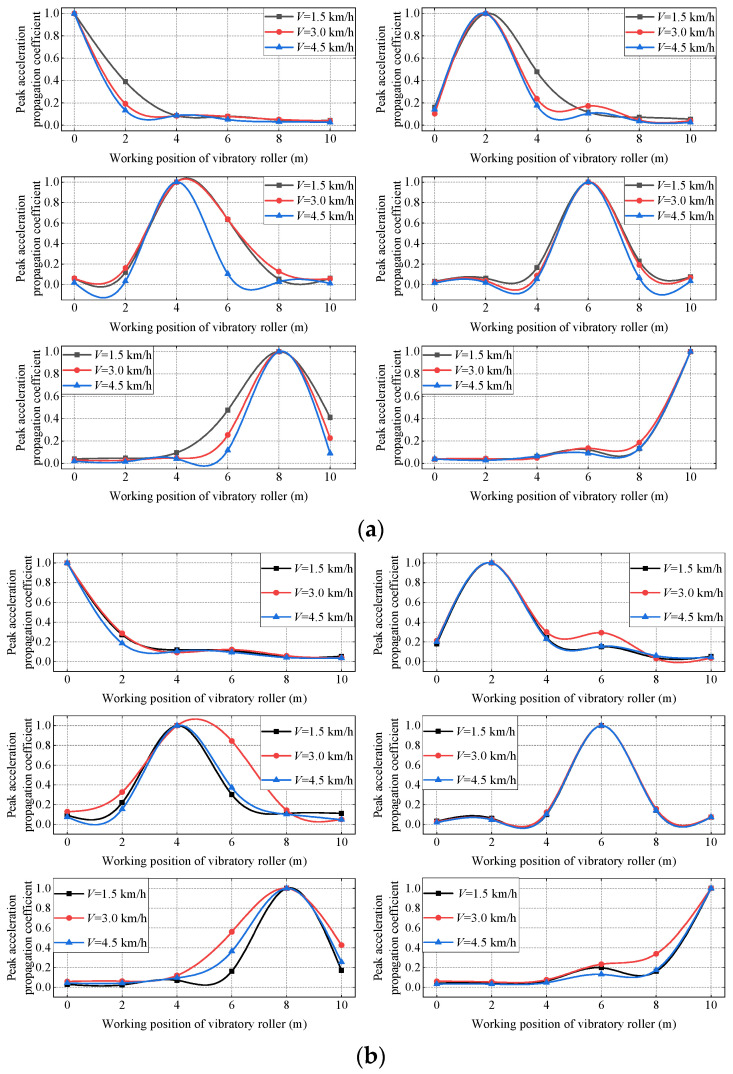
Propagation coefficient of peak acceleration of vibrating roller at different speeds. (**a**) Uncompacted soil. (**b**) Compacted soil.

**Figure 10 sensors-23-02183-f010:**
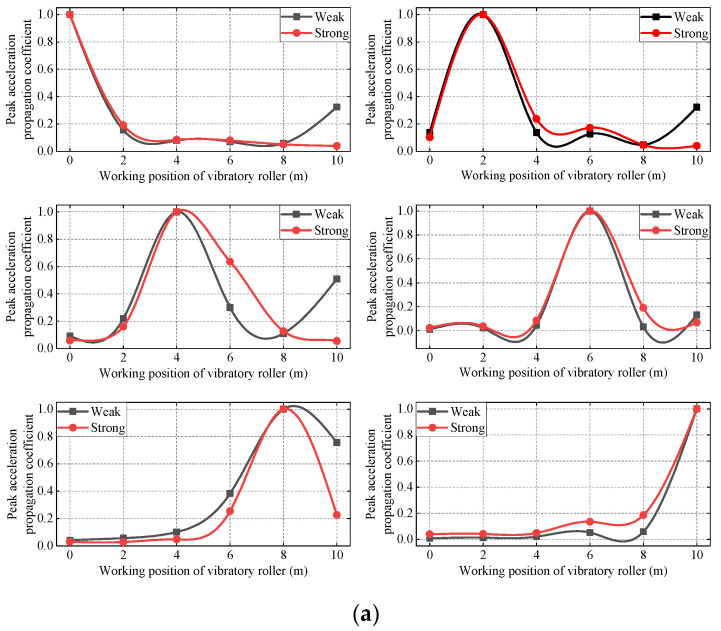
Propagation coefficient of peak acceleration under different amplitudes of vibratory roller. (**a**) Propagation coefficient of peak acceleration under uncompacted state. (**b**) Propagation coefficient of peak acceleration under compacted state.

**Figure 11 sensors-23-02183-f011:**
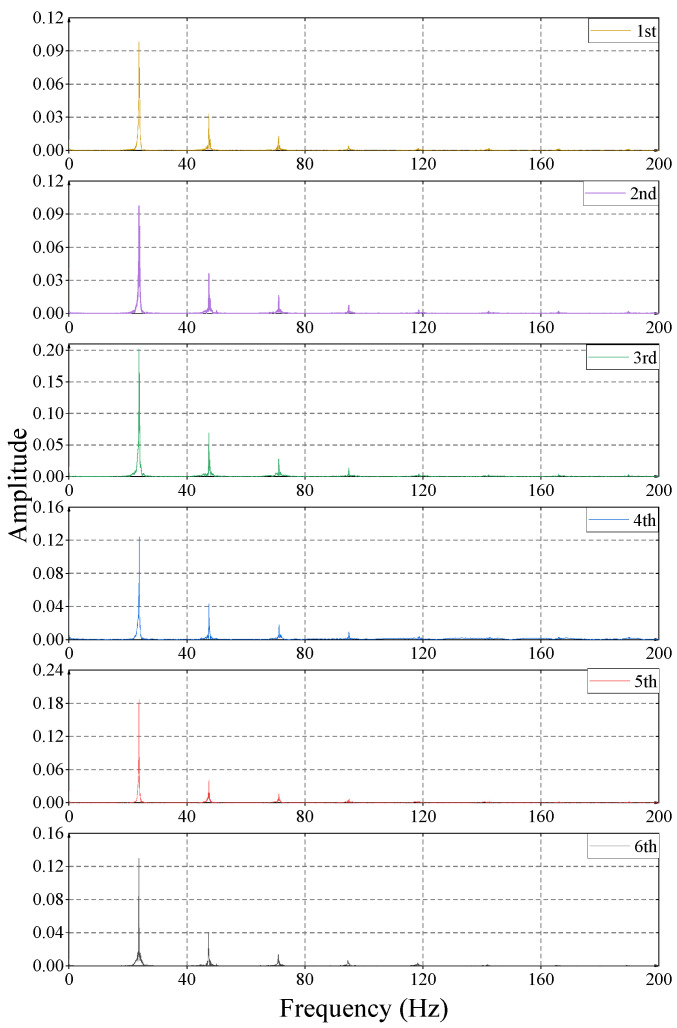
FFT spectrum of the acceleration signal.

**Figure 12 sensors-23-02183-f012:**
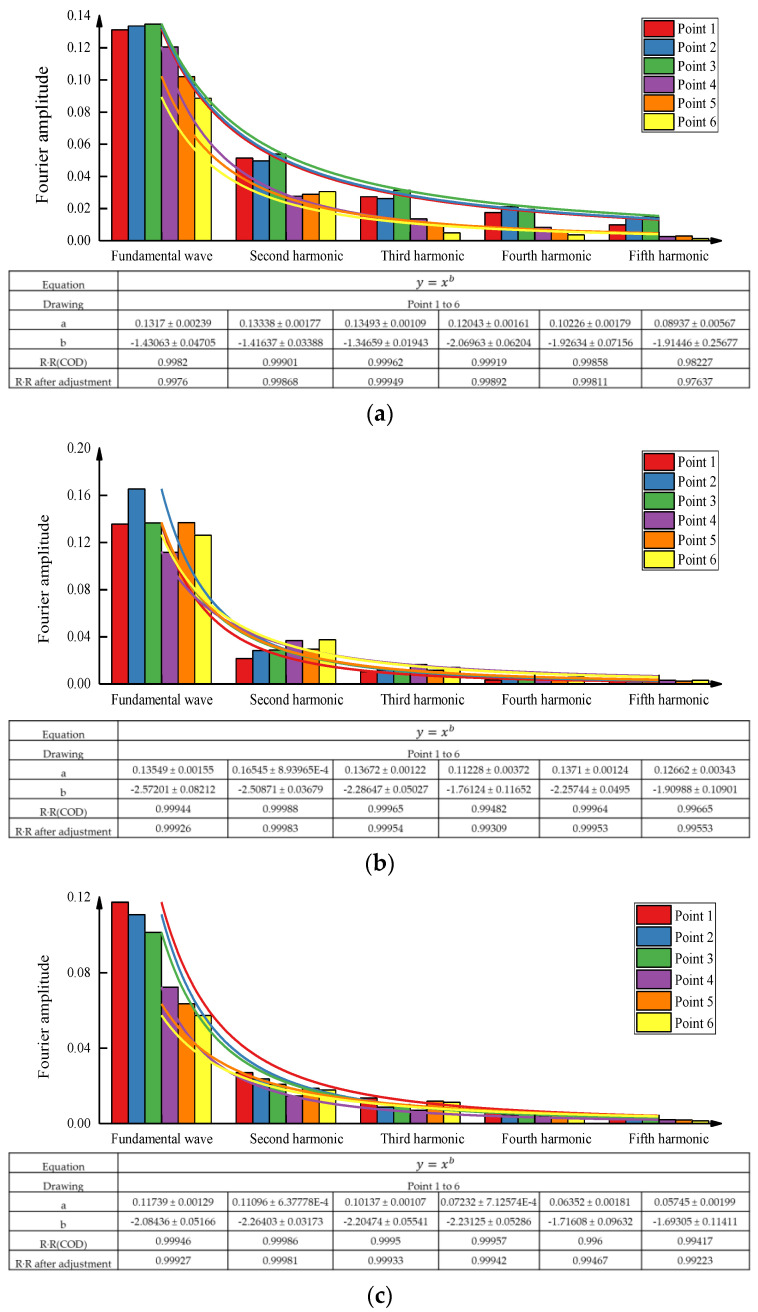
Amplitudes of fundamental and harmonic at different vibratory roller speeds. (**a**) The speed of the vibrated roller is 1.5 km/h. (**b**) The speed of the vibrated roller is 3 km/h. (**c**) The speed of the vibratory roller is 4.5 km/h.

**Figure 13 sensors-23-02183-f013:**
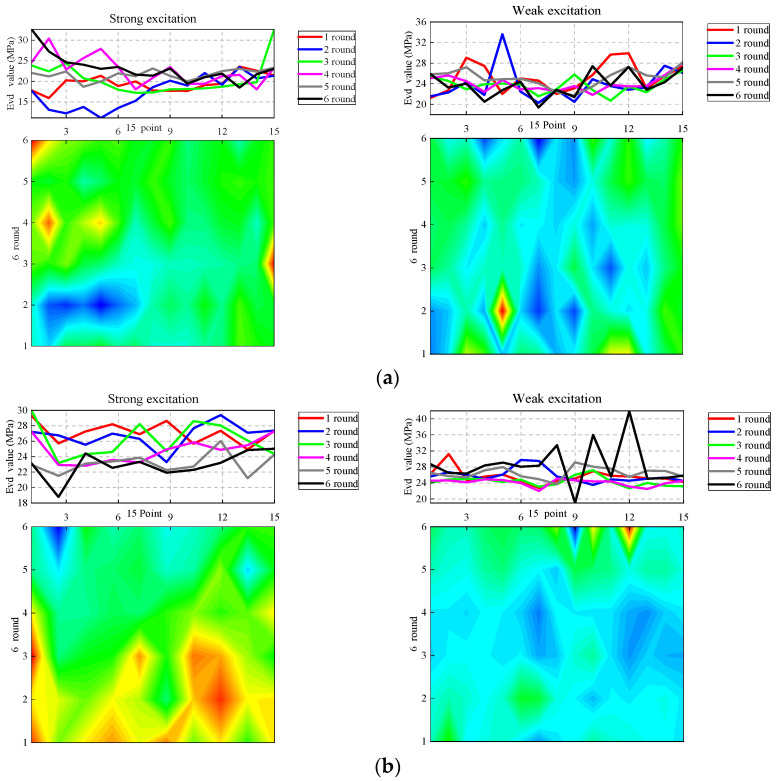
*E*_vd_ contour map. (**a**) The speed of the vibratory roller is 1.5 km/h. (**b**) The speed of the vibrated roller is 3 km/h. (**c**) The speed of the vibrated roller is 4.5 km/h.

**Figure 14 sensors-23-02183-f014:**
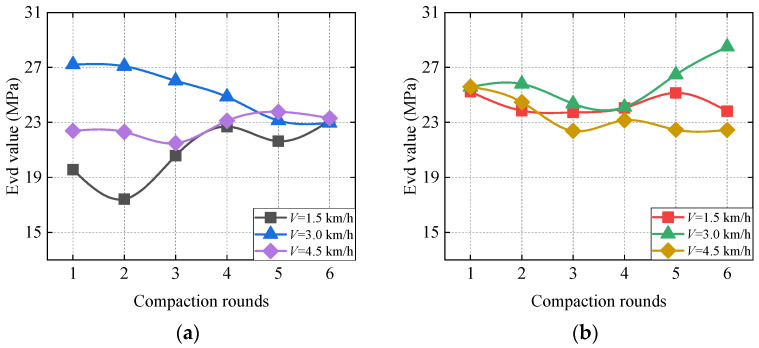
Curve of average *E*_vd_ under different rounds of compaction. (**a**) Strong excitation. (**b**) Weak excitation.

**Table 1 sensors-23-02183-t001:** Vibratory roller parameters.

Parameters	Working Modes	
	Strong Excitation Weak Excitation	
Exciting force (kN)	416	275
Nominal amplitude (mm)	2.05	1.03
Vibration frequency (Hz)	27	31
Mass (kg)	26,700	
Vibration wheel diameter (mm)	1700	
Vibrating wheel width (mm)	2170	

**Table 2 sensors-23-02183-t002:** Test cases.

Case Number	Compaction Rounds	Travel Speed
1	1 static pressure and 6 weak vibration	1.5 km/h
2	1 static pressure and 6 weak vibration	3.0 km/h
3	1 static pressure and 6 weak vibration	4.5 km/h
4	1 static pressure and 6 weak vibration	1.5 km/h
5	1 static pressure and 6 weak vibration	3.0 km/h
6	1 static pressure and 6 weak vibration	4.5 km/h

## Data Availability

The data used in this article are listed in the graphs and tables.

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
