# Peer review of "Wave Propagation Characteristics and Compaction Status of Subgrade during Vibratory Compaction"

_sensors, 2023, doi:10.3390/s23042183_

Round 1

Reviewer 1 Report

This paper studies horizontal direction of the filling soil compaction in the working process of the vibratory roller. The results and analysis are promising and could be useful in the development of engineering construction, I will consider the publication of this article after some revisions.

1. The introduction of this paper only introduces the background and research direction of promotion of vibration wave, and lacks a description of the full paper framework, which is not desirable. The authors should give reasons and make corresponding revisions.

2. What is the meaning of 'Evd'? As an important analysis indicator of this article, there is no explanation or emphasis in the text, and the authors should give a corresponding introduction in the subsequent revision.

3. The description of ' Fig.6 The time history curve of each measuring point ' is not sufficient and the authors should add a proper analysis before drawing a corresponding conclusion.

4. In the Section 4 Conclusion, the fourth conclusion describes too much about the experiment, which appears to be very redundant and not enough to highlight the conclusion. The authors should modify the conclusion to highlight the innovation of the article and the experimental results.

5. This paper still has some questions in English writing, especially in terms of format. For example, case format problem of '; The' in Section 2.1 and 'when' in line 392,the authors should check and revise the whole article.

Author Response

Dear experts

Thank you very much for your suggestions and requests for changes to this paper, I have made the corresponding changes in the text according to each of your comments, and I am here to respond to your five suggestions.

(1)The existing research mainly focuses on the compaction detection index or method, but the research on the propagation law of vibration waves along the horizontal direction is less. In the actual compaction process, the dynamic response characteristics of the vibration wave in the horizontal direction of the filler are also very important. By studying the diffusion law of the vibration wave in the horizontal direction, we can obtain information on the attenuation degree of vibration energy, so as to study the influence of the change of parameters such as travel speed and vibration frequency on the compaction of the filler and improve the compaction efficiency and quality. Based on this, this paper conducts a field prototype test according to the shortage of existing research, obtains the acceleration time range data at different monitoring points during the compaction process by burying acceleration sensors on the surface of the filler, and analyzes the peak propagation coefficient, time history curve and spectral characteristics of the filler surface to study the peak acceleration coefficient and optimum compaction conditions of the filler at different speeds of vibratory rollers. transmission coefficients and optimum compaction conditions at different speeds. It provides some reference for the study of the horizontal compaction of the fill during the operation of the vibratory roller.

(2)The dynamic deformation modulus (Evd) is obtained from a plate test with a certain size and time of action load applied by a falling hammer impact, which represents the ratio of dynamic stress to dynamic strain at a point in the roadbed and describes the ability of the point to resist the dynamic deformation generated by the dynamic load in a certain state.

(3)The acceleration peak at each measurement point can be studied by obtaining the data from the acceleration sensors buried at six measurement points on the surface of the fill, and then the acceleration peak propagation coefficient at the remaining measurement points can be studied when the vibratory roller passes through a certain measurement point. Analysis of this figure shows that the acceleration peak value decreases significantly along the direction of advance of the vibratory roller as the vibration wave in the filler soil.

(4)Here I have trimmed down the original, more lengthy conclusion and reorganized it as: through several rolling tests, it was found that the average Evd value of the vibratory roller under the weak vibration and the speed of 3km/h was significantly higher than the rest of the working conditions.

(5)I have rechecked and revised the text for some formatting issues.

Reviewer 2 Report

This paper does not properly presents its results in the context of international literature but instead, it is concentrated mostly on the Chinese achievements in this area. The Authors should include in the introduction reference to actual practice in Eurocode, ASTM, AASHTO etc. Add discussion of various papers in international journals e.g.: 

https://www.sciencedirect.com/science/article/abs/pii/S2214391218301922

or https://www.sciencedirect.com/science/article/abs/pii/S0013795220305810

And even in this special issue: https://www.sciencedirect.com/journal/transportation-geotechnics/vol/17/part/PB

The last paragraph of the introduction must clearly explain the novelty of the presented results with respect to the existing literature, particularly the above-mentioned references.

Author Response

Dear experts.

Thank you very much for your suggestions and requests for revisions to this paper, and I have revised the paper accordingly. I have revised the paper according to your comments. The revision mainly includes the addition of the deficiencies in the introductory part of the paper, and the addition of the introduction part after referring to the journal articles provided by you, the expert, mainly to understand and describe the international development of road compaction and the compaction quality testing index. The following four articles are cited here.

[1] Badakhshan, E., Noorzad, A., Bouazza, A., et al. Predicting the behavior of unbound granular materials under repeated loads based on the compact shakedown state[J]. Transportation Geotechnics, 2018, 17: 35-47.

[2] Hani, H.T., Habib, T., Ahmed, F., et al. Spatial variability of compacted aggregate bases[J], Transportation Geotechnics, 2018, 17: 56-65.

[3] Jayantha, K., Tanvirul, I., Arooran, S. Review of soil compaction: History and recent developments[J]. Transportation Geotechnics, 2018, 17: 24-34.

[4] Proctor, R.R. Fundamental principles of soil compaction[J]. Engineering News-Record, 1933, 111(9): 55–58.

Reviewer 3 Report

there is no novelty.

Author Response

Dear experts.
Thank you very much for pointing out the problems in this paper. After reviewing the literature and thinking and revising, I have reworked parts of this paper to make appropriate additions and also corrected the formatting errors in the paper.

Reviewer 4 Report

The paper is well written and should be published.

Author Response

Dear experts.
Thank you very much for your approval of this paper. After reviewing the relevant literature, we have made appropriate additions and corrections to parts of this paper, and also corrected the formatting errors in the paper.

Round 2

Reviewer 1 Report

Thank the authors for their efforts. The authors have adequately addressed all my concerns in the review, and did a good job to revise and improve the paper. The paper now is suitable for publication in Sensors in its current form.

Author Response

Dear Experts.

Thank you for your support of this article. I have made minor changes to some of the statements after reading through the whole article, and I will upload the final version after the changes later.

Best regards

Reviewer 2 Report

The paper is suitable for publication, but the English language in many places is awkward.

For example in lines 87-88.  This sentence lacks logic:

(...) The above research mainly focuses on compaction detection index or method, but the research on the propagation law of vibration wave along the horizontal direction is less (...)

There are many more similar language errors. Carry on language editing.

Author Response

Dear experts.

Thank you for your valuable corrections to this article. I have corrected the grammatical errors and contradictory terms in the article after reading through the whole text, and I will upload the revised version soon.

Thank you again for your kind reminder.

Best regards

Reviewer 3 Report

publish

Author Response

(The authors gave the same response as above.)
